# Clinical significance of anti-NT5c1A autoantibody in Korean patients with inflammatory myopathies

**Seung-Ah Lee**[1], **Hyun Joon Lee**[2], **Bum Chun Suh**[3], **Ha Young Shin**[4], **Seung Woo Kim**[4], **Byeol-A Yoon**[5], **Young-Chul Choi**[2,6], **Hyung Jun Park**[2,6]\*

1 Department of Neurology, Mokdong Hospital, Ewha Womans University School of Medicine, Seoul, Republic of Korea, 2 Department of Neurology, Gangnam Severance Hospital, Yonsei University College of Medicine, Seoul, Republic of Korea, 3 Department of Neurology, Kangbuk Samsung Hospital, Sungkyunkwan University School of Medicine, Seoul, Republic of Korea, 4 Department of Neurology, Severance Hospital, Yonsei University College of Medicine, Seoul, Republic of Korea, 5 Department of Neurology, Peripheral Neuropathy Research Center, College of Medicine, Dong-A University, Busan, Republic of Korea, 6 Rehabilitation Institute of Neuromuscular Disease, Gangnam Severance Hospital, Yonsei University College of Medicine, Seoul, Republic of Korea

\* hjpark316@yuhs.ac

## Abstract

To explore the clinical significance of anti-cytosolic 5'-nucleoditase 1A (NT5c1A) antibody seropositivity in inflammatory myopathies, we measured anti-NT5c1A antibodies and analyzed their clinical features. Anti-NT5c1A antibodies were measured in the sera of 103 patients with inflammatory myopathies using an enzyme-linked immunosorbent assay. Positivity for anti-NT5c1A antibody was found in 13 (12.6%) of 103 patients with inflammatory myopathy. Anti-NT5c1A antibody was most frequently identified in patients with inclusion body myositis (IBM) (8/20, 40%), followed by dermatomyositis (2/13, 15.4%), immune-mediated necrotizing myopathy (2/28, 7.1%), and polymyositis (1/42, 2.4%). In eight patients with the anti-NT5c1A antibody-seropositive IBM, the median age at symptom onset was 54 years (interquartile range [IQR]: 48–57 years), and the median disease duration was 34 months (IQR: 24–50 months]. Knee extension weakness was greater than or equal to hip flexion weakness in eight (100%) patients, and finger flexion strength was less than shoulder abduction in three (38%) patients. Dysphagia symptoms were found in three (38%) patients. The median serum CK level was 581 IU/l (IQR: 434–868 IU/L]. Clinically significant differences were not found between anti-NT5c1A antibody-seropositive and seronegative IBM groups with respect to gender, age at symptom onset, age at diagnosis, disease duration, serum CK values, presence of other autoantibodies, dysphagia, and the pattern of muscle impairment. Although anti-NT5c1A antibody is known to be associated with IBM, seropositivity has also been noted in non-IBM inflammatory myopathies, and is insufficient to have clinical significance by itself. These findings have important implications for interpreting anti-NT5c1A antibody test results as the first study in Korea.

**Data Availability Statement:** All relevant data are within the paper and its Supporting information files.

**Funding:** This research was supported by the Basic Science Research Program through the National Research Foundation of Korea (NRF) funded by the Ministry of Education (grant number:2020R1I1A1A01068066) and a new faculty research seed money grant from the Yonsei University College of Medicine for 2020 (6-2020-0127). The funders had no role in study design, data collection and analysis, decision to publish, or preparation of the manuscript.

**Competing interests:** All authors certify that there are no affiliations with or involvement in any organization or entity with direct financial interest in the subject matter or materials discussed in the manuscript. All authors have approved the final version of the manuscript and have no potential conflicts of interest to declare.

## 1. Introduction

Idiopathic inflammatory myopathies are heterogeneous disorders characterized by muscle weakness and inflammation with varying clinical manifestations [1]. Inflammatory myopathies are classified into several subgroups including dermatomyositis, polymyositis, immune-mediated necrotizing myopathy (IMNM), and inclusion body myopathy (IBM) [2]. Over the past decade, the classification criteria for inflammatory myopathies have been significantly updated due to a deeper understanding of the pathogenesis and identification of homogeneous autoantibodies [3].

IBM, a type of inflammatory myopathy, is the most common acquired myopathy affecting patients aged > 50 years [4, 5]. Patients with IBM are clinically characterized by slow progressive muscle weakness that preferentially affects the finger flexors and quadriceps muscles. Although the primary pathogenic mechanism of IBM remains controversial, the pathologic findings of inclusions with autophagic vacuoles and lymphocytic infiltrations have suggested an immune-mediated pathway [5, 6]. In 2013, autoantibodies against cytosolic 5'-nucleotidase 1A (NT5c1A) was identified in the serum of patients with IBM [6]. The NT5c1A is an enzyme that dephosphorylates adenosine monophosphate, which regulate energy balance and muscle contraction [5, 6]. Autoantibodies were expected to serve as biomarkers for IBM diagnosis. However, subsequent studies have shown inconsistent sensitivity for anti-NT5c1A in IBM, ranging 33–76% [6–12]. Additionally, anti-NT5c1A autoantibodies were also detected in patients with other inflammatory myopathies or autoimmune diseases [9, 10, 13]. Therefore, the usefulness of anti-NT5c1A autoantibody as a meaningful biomarker for IBM and its clinical utility needs to be investigated. In Korea, there have been no previous studies on anti-NT5c1A autoantibodies and their clinical associations with inflammatory myopathy, including IBM. Therefore, to understand the clinical utility of the anti-NT5c1A autoantibody in the diagnosis of IBM, we measured the anti-NT5c1A autoantibody in inflammatory myopathies and analyzed the clinicopathological features.

## 2. Materials and methods

We reviewed the medical records between January 2003 and August 2022 in the myopathy database of Gangnam Severance Hospital. In total, 103 patients with inflammatory myopathy were identified. The classification of inflammatory myopathies was established according to the 2017 European League Against Rheumatism/American College of Rheumatology Classification (EULAR/ACR) criteria (with a probability ≥ 55%, corresponding at least to 'probable inflammatory myopathies') [1]. IBM group was defined by new criteria proposed in the EULAR/ACR sub-classification tree based on the clinical phenotypes of finger flexor weakness and no response to treatment or by the status of rimmed vacuoles in histopathologic findings [1]. Since IMNM could not be distinguished from PM in the subclassification tree [1], signal recognition particles (SRP) or 3-hydroxy-3-methylglutaryl-coenzyme A reductase (HMGCR) were included in the diagnostic criteria for IMNM, as proposed based on a recent consensus [14]. Therefore, the 103 inflammatory patients were classified as 20 with IBM, 13 with dermatomyositis, 28 with IMNM, and 42 with polymyositis.

Previously, among noninflammatory muscle diseases, a higher rate compared to other hereditary myopathies of anti-NT5c1A antibody seropositivity was detected in patients with valosin-containing protein (VCP)-related myopathy [13]. We selected four patients with valosin-containing protein (VCP)-related myopathy (S1 Fig). II-1 patient from MF1539, had a heterozygous variant (NM_007126.3: c.463C>T; p.Arg155Cys). Patients II-1 and III-1 from MF1578, had a heterozygous variant (c.476G>A; p.Arg159His). Patient II-1 from MF1952, had a heterozygous variant (c.572G>A; p.Arg191Gln). These variants have been previously

reported to be pathogenic or likely pathogenic variants [15–17]. This study was approved by the Institutional Review Board of Gangnam Severance Hospital, Korea (approval number:3-2022-0365). The requirement for written informed consent was waived by the board because this was a retrospective study.

## 2.1. Clinical and pathologic features of the patients

We retrospectively analyzed clinical, laboratory, and pathologic data through medical record review.

Clinical information used in the phenotype assessment included age at symptom onset, age at diagnosis, disease duration, muscle impairment, dysphagia, and skin lesions. We also assessed serum creatine kinase (CK) levels. Manual muscle testing was performed and was graded using the Medial Medical Research Council (MRC) 5-point scale (0, 1, 2, 3-, 3, 3+, 4-, 4, 4+, 5-, and 5). The MRC grading included shoulder abduction, elbow extension, elbow flexion, wrist extension, wrist flexion, hip flexion, knee extension, knee flexion, ankle dorsiflexion, ankle plantarflexion, and neck flexion. Especially for IBM group, muscle impairment pattern was checked whether there was muscle weakness in the knee extensors and finger flexors compared to adjacent muscles.

Pathological findings were retrieved from previous pathology reports. Histological reports of patients with IBM were analyzed, and the records were available for 16 of the 20 patients. Frozen 10 μm-thick muscle tissue sections were studied after staining with hematoxylin and eosin, modified Gomori trichrome, and nicotinamide adenine dinucleotide–tetrazolium reductase. Neuromuscular specialists and pathologists assessed the samples. We recorded histopathological features, including the presence of rimmed vacuoles (RV) and endomysial inflammatory infiltrates surrounding or invading nonnecrotic muscle fibers. In addition to the IBM group, other inflammatory myopathy patients with anti-NT5c1A seropositivity were further analyzed for pathology, if available (P9, P10, P12, P13).

Four patients (P1, P2, P6, and P7) underwent lower-limb magnetic resonance imaging (MRI) of the pelvis, thigh, and calf muscles on a 3.0T system (MAGNETOM Vida, Siemens, Erlangen, Germany). The imaging was performed in the axial (field of view [FOV], 24–32 cm; slice thickness, 10mm; slice gap 0.5–1mm) and coronal planes (FOV, 38–40cm; slice thickness, 4–5mm; slice gap 0.5–1.0 mm). The protocol used for all patients was T1-weighted spin-echo (SE) (repetition time [TR] 570–650ms, echo time [TE] 14–20ms, 12 metrics) and short tau inversion recovery (STIR) -weighted SE (TR 3,090–4,900ms, TE 85–99ms, and 12 metrices).

## 2.2. Serological testing for autoantibodies

Testing for anti-NT5c1A autoantibodies was performed in the sera of 103 patients with inflammatory myopathy and four VCP-related myopathies. The anti-NT5c1A autoantibodies were identified using an enzyme-linked immunosorbent assay (ELISA) (Euroimmune, Lubeck, Germany). Microplates coated with recombinant NT5c1A were incubated with the diluted serum. Anti- NT5c1A antibodies were measured using a previously reported ELISA method according to the manufacturer's standard protocol [4]. The results were evaluated semiquantitatively as a ratio (optical density [OD] at 450 nm of the sample/$OD_{450}$ of the calibrator [cutoff]); a ratio of $\geq 1$ was deemed positive.

Autoantibodies against myositis-specific autoantibodies were tested using a line blot immunoassay of Euroline Autoimmune Inflammatory Myopathies 16 Ag (Euroimmune, Lubeck, Germany). The test kit contained 16 antigens, including Mi-2α, Mi-2β, TIF1γ, MDA5, NXP2, SAE1, Ku, PM-Scl100, PM-Scl75, Jo-1, SRP, PL-7, PL-12, EJ, OJ, and Ro-52. The results of more than medium-to-strong bands were considered positive.

## 2.3. Statistical analysis

The results of anti-NT5c1A antibody-positive and -negative patients with IBM were compared. In addition, we compared the IBM and non-IBM inflammatory myopathies among anti-NT5c1A antibody-positive patients. Statistical analysis was performed using R software (version 4.2.0, www.r-project.org) to analyze the results. Fischer's exact test was used to compare categorical variables. Age at symptom onset, disease duration, age at diagnosis, and serum CK level were analyzed using the Mann–Whitney test. Statistical significance was set at $p \leq 0.05$.

# 3. Results

## 3.1. Demographic data

Table 1 summarize the demographic finding of 103 inflammatory myopathy patients. There were 40 males (39%) and 63 females (61%). The median ages at symptom onset and diagnosis were 55 (IQR: 44–63 years) and 57 (IQR: 57–65) years. The median disease duration was 10 (IQR: 3–32) months. Median serum CK levels were 2,777 (IQR: 545–7,231), the highest at 7,171 (IQR: 4,214–11,310) in IMNM, followed by polymyositis, dermatomyositis, and IBM.

## 3.2. Anti-NT5c1A antibody ELISA

The anti-NT5c1A antibody values are shown in Fig 1. Positivity for anti-NT5c1A antibody was found in 13 (12.6%) of 103 patients with inflammatory myopathy. In the inflammatory myopathy group, anti-NT5c1A antibody was most frequently identified in patients with IBM (8/20, 40%), followed by dermatomyositis (2/13, 15.4%), IMNM (2/28, 7.1%), and polymyositis (1/42, 2.4%). Relatively high titer of anti-NT5c1A antibody (OD ratio≥3) was only confirmed in the IBM group. The anti-NT5c1A antibody was detected in one patient with VCP-related myopathy (III-1 patients from MF1578, see S1 Fig) without showing any relevant clinical symptoms, including muscle weakness or dementia. However, the other three patients with muscle weakness or dementia tested negative for anti-NT5c1A antibody. A summary of the clinicopathological characteristics of the patients with anti-NT5c1A antibodies is shown in Table 2.

## 3.3. Clinicopathological features of anti-NT5c1A antibody seropositive patients with IBM

Among 20 patients with IBM, eight (40%) were seropositive for the anti-NT5c1A antibody. In the anti-NT5c1A antibody-seropositive group, the median age at symptom onset was 54 years

**Table 1. Demographic characteristics of 103 patients with inflammatory myopathies.**

|  | Total patients | IBM | Dermatomyositis | IMNM | Polymyositis |
|---|---|---|---|---|---|
| Number of patients | 103 | 20 (19%) | 13(13%) | 28 (27%) | 42 (41%) |
| Age at onset, Y | 55 [44–63] | 56 [50–62] | 43 [35–52] | 54 [45–66] | 56 [48–61] |
| Age at diagnosis, Y | 57 [57–65] | 59 [55–65] | 44 [36–56] | 56 [45–65] | 57 [49–64] |
| Duration, M | 10 [3–32] | 36 [24–50] | 6 [4–12] | 6 [3–14] | 7 [3–12] |
| Males | 40 (39%) | 10 (50%) | 4 (31%) | 9 (32%) | 17 (40%) |
| CK (IU/L) | 2,277 [545–7,231] | 530 [315–833] | 1,151 [90–3914] | 7,174 [4,214–11,310] | 2,321 [526–7,005] |

Note: Values are expressed as number (%) or median [interquartile range].

IBM, inclusion body myositis; IMNM, immune-mediated necrotizing myopathy

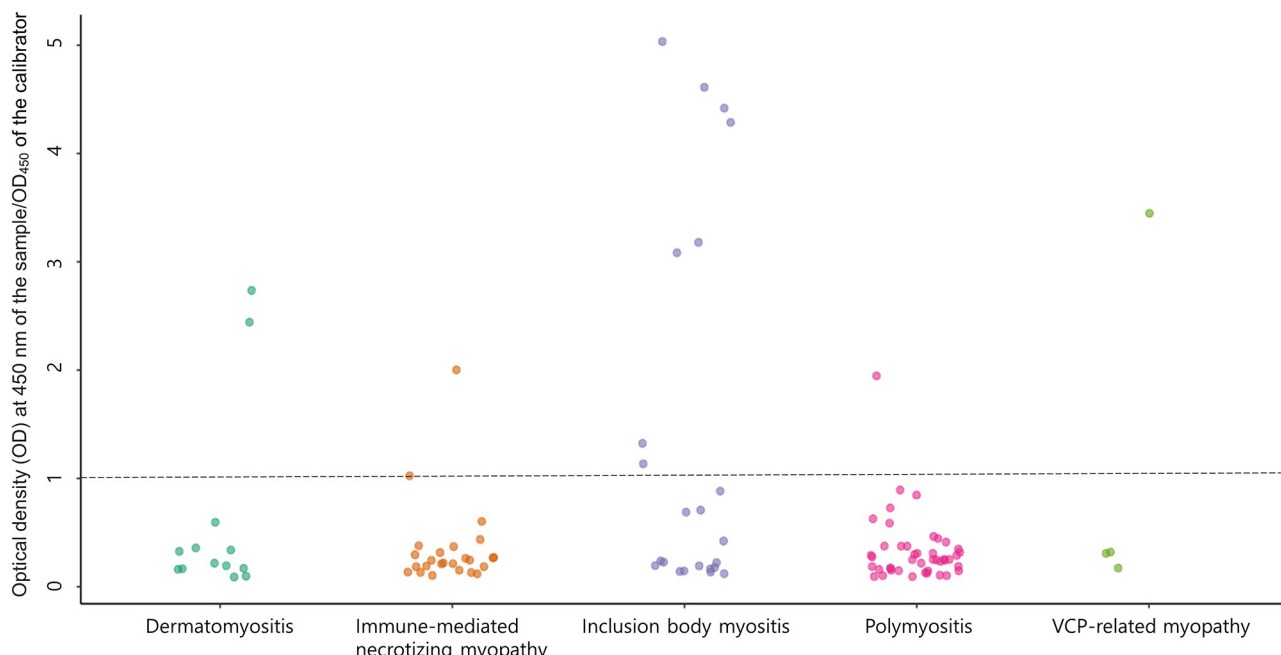

**Fig 1. Anti-cytosolic 5'-nucleotidase 1A (NT5c1A) antibody ELISA.** Antibodies reactive with recombinant NT5c1A protein by ELISA in sera from patients with inflammatory myopathy and valosin-containing protein (VCP)-related patients with myopathy. The cut-off level for positivity is indicated by the broken line.

**Table 2. Clinicopathologic presentation and laboratory parameters of patients with anti-NT5C1A antibody (n = 14).**

| Clinical diagnosis | Inclusion body myositis | | | | | | | | Dermatomyositis | | IMNM | PM | | VCP disease |
|---|---|---|---|---|---|---|---|---|---|---|---|---|---|---|
| Patient | P1 | P2 | P3 | P4 | P5 | P6 | P7 | P8 | P9 | P10 | P12 | P11 | P13 | P14 |
| Age at onset, Y | 51 | 56 | 40 | 37 | 64 | 52 | 57 | 57 | 39 | 25 | 44 | 60 | 76 | - |
| Age at diagnosis, Y | 54 | 59 | 61 | 39 | 66 | 57 | 59 | 61 | 39 | 25 | 46 | 60 | 76 | 29 |
| Gender | M | F | M | M | F | F | M | M | M | F | M | F | F | F |
| Duration, M | 32 | 36 | 252 | 24 | 24 | 55 | 24 | 48 | 1 | 1 | 18 | 1 | 3 | - |
| Dysphagia | - | + | - | - | - | - | + | + | - | - | - | - | - | - |
| Skin lesion | - | - | - | - | - | - | - | - | + | + | - | - | - | - |
| CK (IU/L) | 652 | 270 | 224 | 488 | 978 | 831 | 509 | 1234 | 27217 | 52 | 4197 | 1200 | 15 | n/a |
| Other antibodies* | - | Ro-52 (+++) | Ro-52 (+++) | - | SRP (+) | - | - | MDA5 (+) SAE1(+) | - | MDA5 (+++) PM-Scl75 (+) PL-7(+) | SRP (+++) OJ(+) SAE1 (+) | HMGCR | PL-7 (+) | - |
| Muscle biopsy | | | | | | | | | | | | | | |
| Inclusion bodies | + | + | + | + | - | - | n/a | n/a | - | - | - | n/a | - | n/a |
| Necrosis fibers | Many | Few | - | Few | Many | Many | n/a | n/a | - | - | - | n/a | Few | n/a |
| perimysium cell infiltrate | + | + | + | - | + | + | n/a | n/a | - | - | - | n/a | + | n/a |

IMNM, immune-mediated necrotizing myopathy; PM, polymyositis; Y, years; M, months; F, female; M, male;

*Antibodies against HMGCR, Mi-2α, Mi-2β, TIF1γ, MDA5, NXP2, SAE1, Ku, PM-Scl100, PM-Scl75, Jo-, SRP, PL-7, PL-12, EJ, OJ, and Ro-52 are included, +, medium band; ++, strong band; +++, very strong band

**Table 3. Clinicopathological features of 20 patients with inclusion body myositis according to anti-NT5c1A antibody status.**

| | Patients with inclusion body myositis (n = 20) | | |
| --- | --- | --- | --- |
| | anti-NT5c1A positive (n = 8) | anti-NT5c1A negative (n = 12) | p-value |
| Male | 5 (63) | 5 (42) | 0.650 |
| Age at symptom onset (Y) | 54 [48–57] | 56 [50–62] | 0.384 |
| Age at diagnosis (Y) | 59 [56–61] | 60 [54–66] | 0.628 |
| Disease duration (M) | 34 [24–50] | 48 [32–66] | 0.678 |
| Serum CK (IU/l) | 581 [434–868] | 453 [240–766] | 0.492 |
| Other antibodies* | 4 (50) | 3 (25) | 0.356 |
| Dysphagia | 3 (38) | 6 (50) | 0.670 |
| Knee extension weakness (≥ hip flexion weakness) | 8 (100) | 8 (67) | 0.117 |
| Finger flexion weakness (> shoulder abduction weakness) | 3 (38) | 8 (67) | 0.362 |
| Rimmed vacuoles | 4/6 (67) | 3/10 (30) | 0.302 |
| Endomysial inflammatory infiltrate | 6/6 (100) | 9/10 (90) | 1.000 |

Note: Values are expressed as number (%) or median [interquartile range].

*Antibodies against Mi-2α, Mi-2β, TIF1γ, MDA5, NXP2, SAE1, Ku, PM-Scl100, PM-Scl75, Jo-1, SRP, PL-7, PL-12, EJ, OJ, and Ro-52 are included.

(IQR: 48–57 years), and the median disease duration was 34 months (IQR: 24–50 months). Knee extension weakness was greater than or equal to hip flexion weakness in eight (100%) patients, and finger flexion strength was less than shoulder abduction in three (38%) patients. Dysphagia symptoms were found in three (38%) patients. The median serum CK level was 581 IU/l (IQR: 434–868 IU/L). We compared the clinicopathological characteristics according to the anti-NT5c1A antibody status (Table 3). No significant differences were detected with regard to sex, age at symptom onset, age at diagnosis, disease duration, serum CK values, status of other autoantibodies, dysphagia, and the pattern of muscle impairment. The 16 available muscle specimens were subjected to pathological analysis: 6 in the anti-NT5c1A antibody seropositive group and 10 in the anti-NT5c1A antibody seronegative group. RV was present in four out of six (67%) patients with anti-NT5c1A antibody seropositivity and in three out of ten (30%) patients with anti-NT5c1A antibody seronegativity. Endomysial inflammation was found in all six patients who were seropositive for anti-NT5c1A and 9 out of 10 (90%) patients in the seronegative group (Fig 2 Muscles from P4 and P5). However, no pathologic findings seemed to correlate with the seropositivity of anti-NT5c1A antibody in IBM (RV, p = 0.302; endomysial inflammatory infiltrate, p = 1.000).

In T1-weighted lower-limb MR images, muscle atrophy and moderate-to-severe fatty replacement of the quadriceps muscles were found in three patients (P1, P2, and P7) (Fig 3). At the pelvis level (A, G, M, and S), the iliopsoas muscles (*) were almost normal even at advanced stages without any inflammatory changes (D, J, P, and V). At the thigh level (B, H, N, and T), muscle atrophy and moderate-to-severe fatty replacement of the quadriceps(arrowheads) muscles were observed in three patients (B, N, and T). At the calf level (C, I, O, and U), the medial gastrocnemius (white arrows) muscles were involved at the initial stage and the fatty replacement of tibialis anterior, extensor digitorum, and peroneus muscles at the advanced stage (O and U). However, the adductor longus and magnus muscles showed more prominent fatty replacement than did the quadriceps muscles in P6 patients (H).

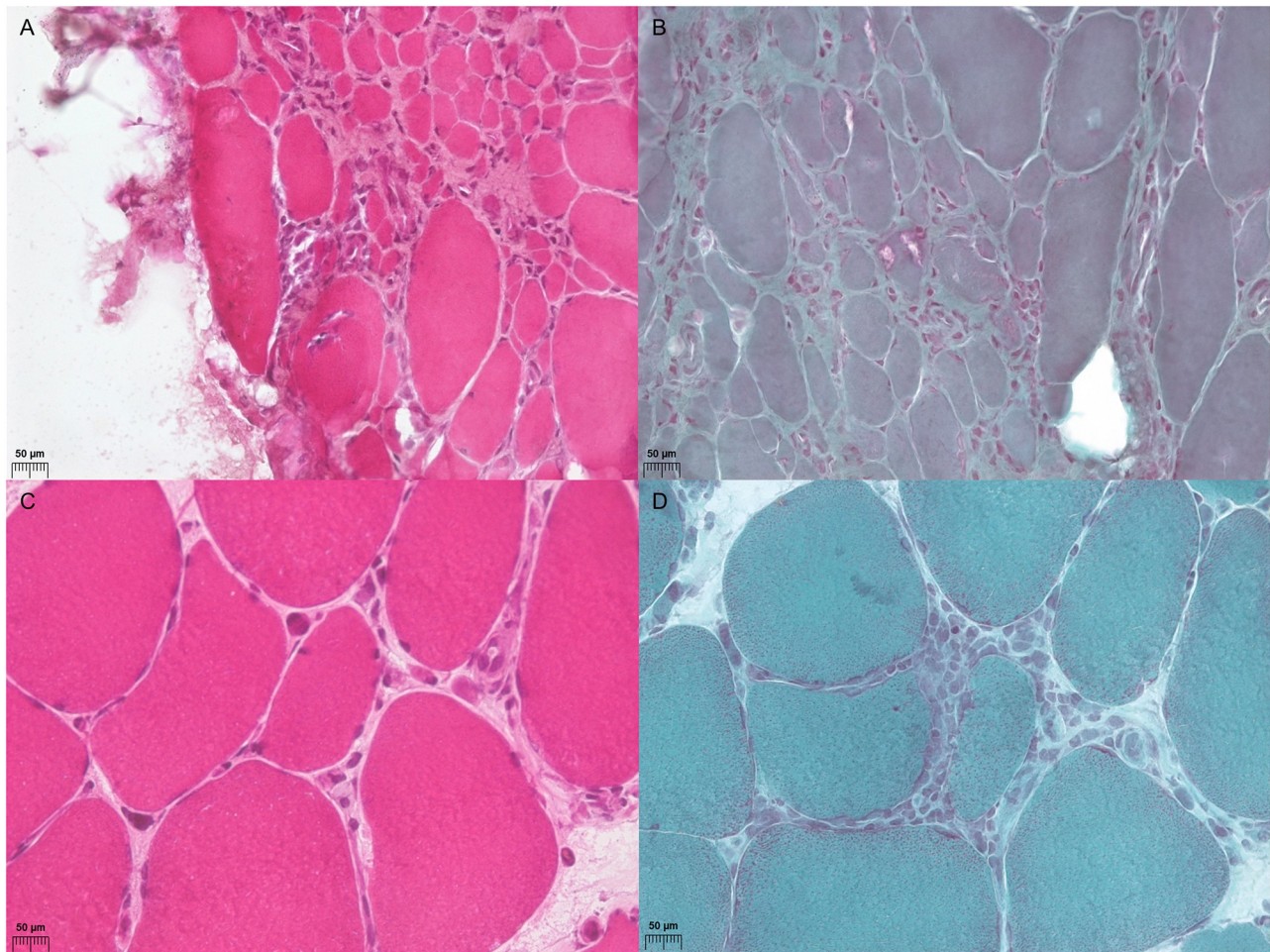

**Fig 2.** Pathology findings in patients with anti-NT5c1A antibody seropositive IBM: P4 (A and B) and P5 (C and D). In patient P4, pronounced muscle fiber size variation with atrophic changes and internalized nuclei, endomysial inflammatory infiltrates, and rimmed vacuoles. (A, hematoxylin and eosin [H&E] stain, B, modified Gomori trichrome [GT] stain, Magnification: x400). In patient P5, variable sizes of muscle fibers with endomysial inflammatory infiltrate without any partial muscle fiber invasion. (C, H&E stain, D, modified GT stain, Magnification: x400).

Short tau inversion recovery (STIR) signal increase was patchy and asymmetric in the thigh and calf muscles. At the pelvis level (D, J, P, and V), there were not high-signal changes in four patients. At the thigh level (E, K, Q, and W), hypersignals of the vastus lateralis, medialis, and intermedius muscles were identified in three patients (P1, P2, and P7). At the calf level, the anterior calf muscles showed no signal changes during the early stages (F and L).

Testing for myositis-related autoantibodies in the IBM group revealed reactivity against Ro-52 in 4 patients. Reactivity against SRP, MDA5, SAE1, and Mi-2β were identified once each. Anti-SRP (P5), anti-MDA5 (P8), and anti-SAE1(P8) autoantibodies were found only in the anti-NT5c1A antibody seropositive group with IBM (Table 2 and S1 Table).

### 3.4. Clinical features of anti-NT5c1A antibody seropositive other inflammatory myopathies

Among the 13 patients with inflammatory myopathy who were seropositive for anti-NT5c1A, five were non-IBM patients (2 dermatomyositis, 2 IMNM, and 1 polymyositis). In this group, the median age at symptom onset was 44 years (IQR: 39–60 years), and the median disease

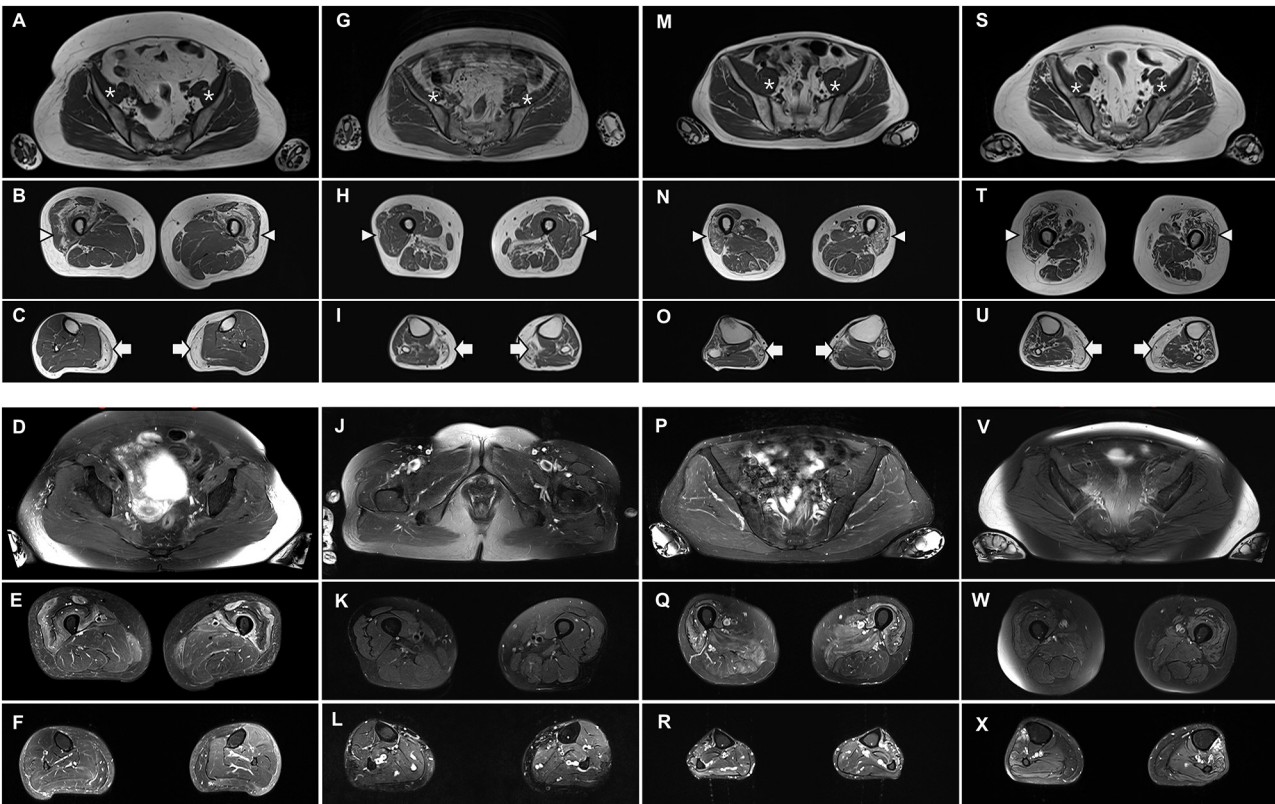

**Fig 3. Lower limb magnetic resonance images of four inclusion body from patients with myositis and seropositive for anti-NT5c1A antibody: P2 (A-F), P6(G-L), P7(M-R), and P1 (S-X) taken 45, 55, 70, and 84 months after symptom onset, respectively.**

duration was 1 month (IQR: 1–3 months). Dysphagia symptoms were not found in any patient, and skin rash was noted in two patients with dermatomyositis. The median serum CK level was 1,200 IU/l (IQR: 52–4,197 IU/L). We compared the clinical characteristics of the anti-NT5c1A antibody-seropositive IBM group (Table 4). Except for the short disease duration in the non-IBM group (p = 0.002), no significant differences were found.

**Table 4. Comparison of clinical features between inclusion body myositis (IBM) and non-IBM inflammatory myopathies with anti-NT5c1A antibody.**

|  | Patients with anti-NT5c1A (n = 13) | | |
|---|---|---|---|
|  | **Inclusion body myositis (n = 8)** | **Other inflammatory myopathies (n = 5)** | **p-value** |
| Male | 5 (63) | 2 (40) | 0.592 |
| Age at symptom onset (Y) | 54 [48–57] | 44 [39–60] | 0.833 |
| Age at diagnosis (Y) | 59 [56–61] | 46 [39–60] | 0.435 |
| Disease duration (M) | 34 [24–50] | 1 [1–3] | 0.002 |
| Serum CK (IU/l) | 581 [434–868] | 1,200 [52–4,197] | 0.661 |
| Dysphagia | 3 (38) | 0 (0) | 0.231 |
| Skin rash | 1 (13) | 2 (40) | 0.510 |
| Other antibodies* | 4 (50) | 4 (80) | 0.565 |

Note: Values are expressed as number (%) or median [interquartile range]. Y, years; M, months; CK, creatine kinase.

*Antibodies against Mi-2α, Mi-2β, TIF1γ, MDA5, NXP2, SAE1, Ku, PM-Scl100, PM-Scl75, Jo-1, SRP, PL-7, PL-12, EJ, OJ, and Ro-52 are included.

Among the 103 patients with inflammatory myopathy, we compared the anti-NT5c1A antibody-seropositive and -seronegative patients (S2 Table). The median serum CK level was 652 IU/l (IQR: 270–1,200) in patients with seropositive inflammatory myopathy and 2,921 IU/l (IQR: 716–7,801) in those with seronegative inflammatory myopathy (p = 0.018). Except for the CK value, no significant differences were detected in sex, age at symptom onset, age at diagnosis, disease duration, dysphagia symptoms, skin rash, or other autoantibody status. The same statistical analysis was performed, with the exception of 20 patients (S3 Table). No statistically significant difference in any clinical feature, including serum CK levels (p = 0.284) was detected. The disease duration was shorter with the exception of patients with IBM; however, no significant differences were observed between the two groups.

## 4. Discussion

In this study, we measured anti-NT5c1A antibody levels in 103 patients with inflammation and found seropositivity in 13 of 103 (12.6%) patients. The seropositivity of anti-NT5c1A antibody was 40% sensitive for the diagnosis of IBM. Compared with the results of previous studies that suggested a sensitivity range of 33–76% [5–7, 9, 18], this result is slightly low. The reason for the variable seropositivity might be attributed to the different techniques used in different centers, different cut-off values used to determine positivity, or different diagnostic criteria in patient selection. When only the results of previous studies using the ELISA method were compared, the sensitivity for IBM diagnosis was almost similar [4, 9, 11, 19].

We also confirmed that anti-NT5c1A antibodies were identified in the patients with dermatomyositis (2/13), IMNM (2/28), and polymyositis (1/42). This finding is consistent with those of other studies that detected anti-NT5c1A antibodies in other inflammatory myopathies. In dermatomyositis or IMNM, the seropositivity of anti-NT5c1A antibody was the second highest after IBM, but the lowest detection rate was observed in polymyositis [4, 6, 10, 11, 13, 18]. This consistency suggests that anti-NT5c1A antibody testing may be helpful in distinguishing polymyositis from IBM when clinicopathological findings are not sufficient to differentiate between the two groups. This distinction is important because polymyositis and IBM respond differently to immunosuppressive therapies. In addition, typical histological abnormalities of IBM are not in all muscle pathologies of patients. Many IBM patients have been initially misdiagnosed with polymyositis [20]. When comparing the clinical findings between the anti-NT5c1A antibody seropositive IBM group and non-IBM group, disease duration was shorter in the non-IBM group than in the IBM group. This is probably due to the nature of the disease course of IBM, which progresses more slowly than dermatomyositis, IMNM, and polymyositis and is not an anti-NT5c1A antibody-related feature. However, it needs to be interpreted taking into account the small number and heterogeneity of non-IBM group.

Previous studies have suggested that seropositivity for anti-NT5c1A antibodies could be associated with a severe phenotype with dysphagia and a high mortality risk [8, 12]. However, in this study, no significant disease characteristics were detected including the distribution of weakness and dysphagia. To date, no clear clinical association exists between anti-NT5c1A antibody-positive and -negative patients in IBM [5, 11–13]. No definite evidence for the clinical association between anti-NT5c1A antibody and specific clinical presentation has yet been reported.

In addition to the current study, several studies have evaluated the correlation of anti-NT5c1A antibody with pathological findings in IBM [5, 10–13]. Several studies demonstrated no relevance between anti-NT5c1A antibody seropositivity and the presence of rimmed vacuoles [5, 11–13], consistent with our findings. However, there is also a report that anti-NT5c1A antibody seropositive IBM group represented with a lower frequency of rimmed vacuoles than

seronegative IBM group [10]. These inconsistent findings in pathological observation are difficult to explain and may be dependent on the different sample sizes, assay method used in individual centers, and the time lapse between symptom onset and pathological examination.

Muscle MRI findings in IBM are known to have preferential involvement, mainly in the flexor digitorum profundus, anterior compartment of distal thigh muscles, and medial gastrocnemius, while the tibialis posterior, soleus, and pelvic muscles are known to be relatively preserved [20–22]. In our patients with IBM, the most affected muscles were the quadriceps and medial gastrocnemius muscles of the lower extremities. One patient had remarkable fat infiltration in the bilateral adductor muscles (adductor longus and magnus) with mild atrophy in the rectus femoris muscles. This finding is contrary to a previous study, in which the adductor longus and adductor magnus were most frequently spared in IBM [21]. Interestingly, a recent study revealed that the distal part of the adductor muscles infiltrated in more than 85% of patients with IBM, considering the proximal to distal gradient [20]. Muscle edema with inflammation is frequently observed in IBM, and signal changes tend to be heterogeneous and asymmetric [23]. In our study, all four patients presented with a high STIR signal. The pattern of inflammation tends to be more remarkable in the anterior muscle of the thigh and lower legs and asymmetric, consistent with reports of previous studies [23]. Muscle MRI scan was performed and analyzed only in some patients with IBM who were positive for anti-NT5c1A antibody, which was insufficient to characterize seropositive patients with IBM. No comparative studies have been conducted on muscle MR imaging to analyze the status of the anti-NT5c1A antibody in IBM.

No significant difference was detected in myositis-specific autoantibody testing with regard to reactivity to other autoantibodies in anti-NT5c1A antibody positive patients with IBM, consistent with previous publications [4, 13]. Antibodies against Ro-52 were most frequently detected in patients with IBM. However, this result must be considered with caution due to the small sample size and the known ubiquitous reactivity of anti-Ro52 autoantibodies [24, 25]. Owing to the discovery of concurrent identification of anti-Ro-52 and anti-NT5c1A antibodies, a hypothesis has been proposed regarding the pathophysiological relationship between the two in patients with IBM; however, this was insufficient to prove an association [4].

In this study, seropositivity for anti-NT5c1A antibody was detected in one of the four patients carrying pathogenic variants of *VCP*. However, the only patient with anti-NT5c1A seropositivity did not show any clinical symptoms. There was a previous study to represent that the seropositivity for anti-NT5c1A antibody was higher among noninflammatory myopathies [13]. In addition, pathogenic variants of *VCP* have been identified in a small number of IBM patients [26, 27]. Along with previous findings, the result may support the possibility of a shared pathway with IBM and VCP-related myopathy in the production of anti-NT5c1A antibodies [13]. However, this is a still a rare finding and requires caution in interpretation. To date, no association has been established between VCP-related myopathy, anti-NT5c1A antibody, and IBM.

The current study had several limitations. First, the clinical information was collected retrospectively, resulting in insufficient or missing data, and was collected for approximately 20 years, with potential chronology bias. Muscle weakness was evaluated with the MRC grade in our study. Therefore, further prospective studies are needed using validated/standardized clinical scales such as the adult myopathy assessment tool. Second, a widely determined gold standard is yet to be established for detecting anti-NT5c1A antibodies. Third, the number of patients was relatively small and clinically relevant correlations may not have been found. Therefore, further prospective studies with larger numbers of patients are needed to provide more insight into the clinical implications of anti-NT5c1A antibody testing.

In conclusion, this study is the first to detect anti-NT5c1A antibodies in various inflammatory myopathies, in addition to IBM in Korea. Seropositivity can be observed in other inflammatory myopathies besides IBM and can be used to differentiate IBM from polymyositis. We also provide the clinical, pathological, and radiological features of Korean patients with IBM using the anti-NT5c1A antibody. We expect that this information would be used as basic data to conduct further studies on IBM and related biomarkers. In addition, no definite clinico-pathological differences are present according to the status of anti-NT5c1A antibody testing. Finally, clinicians should interpret the results of anti-NT5c1A antibody testing along with thorough neurological examinations and pathologic findings.

## Supporting information

**S1 Fig. Pedigree of four patients from three unrelated familied with carrying *VCP* pathogenic variants.**
(TIF)

**S1 Table. Reactivity of myositis-specific autoantibodies in sera from patients with IBM.**
(DOCX)

**S2 Table. Clinical features of 103 inflammatory myopathy patients according to anti-NT5c1A antibody status.**
(DOCX)

**S3 Table. Clinical features of patients with inflammatory myopathies other than IBM according to anti-NT5c1A antibody status.**
(DOCX)

## Acknowledgments

The authors would like to thank the patients for their assistance with this study.

## Author Contributions

**Conceptualization:** Seung-Ah Lee, Young-Chul Choi, Hyung Jun Park.

**Data curation:** Seung-Ah Lee, Hyun Joon Lee, Bum Chun Suh, Ha Young Shin, Seung Woo Kim, Byeol-A Yoon, Young-Chul Choi, Hyung Jun Park.

**Formal analysis:** Seung-Ah Lee, Hyung Jun Park.

**Funding acquisition:** Hyung Jun Park.

**Methodology:** Seung-Ah Lee, Hyun Joon Lee, Byeol-A Yoon.

**Supervision:** Young-Chul Choi.

**Writing – original draft:** Seung-Ah Lee, Hyung Jun Park.

**Writing – review & editing:** Seung-Ah Lee, Young-Chul Choi, Hyung Jun Park.

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
