## [Decision Letter · Decision Letter 0]

27 Feb 2023

PONE-D-22-33685

Clinical significance of anti-NT5c1A autoantibody in Korean patients with inflammatory myopathies

PLOS ONE

Dear Dr. LEE,

Thank you for submitting your manuscript to PLOS ONE. After careful consideration, we feel that it has merit but does not fully meet PLOS ONE’s publication criteria as it currently stands. Therefore, we invite you to submit a revised version of the manuscript that addresses the points raised during the review process.

Major revision

We look forward to receiving your revised manuscript.

Kind regards,

Latika Gupta

Academic Editor

PLOS ONE

Journal Requirements:

“This research was supported by the Basic Science Research Program through the National Research Foundation of Korea (NRF) funded by the Ministry of Education (grant number:2020R1I1A1A01068066)”

Additional Editor Comments:

Please address reviewer comments as appended.

Reviewers' comments:

Reviewer's Responses to Questions

**Comments to the Author**

1. Is the manuscript technically sound, and do the data support the conclusions?

Reviewer #1: Partly

2. Has the statistical analysis been performed appropriately and rigorously? 

Reviewer #1: Yes

3. Have the authors made all data underlying the findings in their manuscript fully available?

Reviewer #1: Yes

4. Is the manuscript presented in an intelligible fashion and written in standard English?

Reviewer #1: Yes

5. Review Comments to the Author

Reviewer #1: I have read with interest the work of Lee et al.

The paperi s well written and the main problem is the lack of novelty.

These are my comments:

Abstracts:

Lines 42-43: in the abstract the authors presented only one group (inflammatory myopathies). The expression "In the inflammatory myopathy group" is therefore superfluous.

Line 51: “no clinically significant difference”. What are the clinical scales used?

Introduction:

Lines 75-77: “to date, no other studies….in Korea”. Rephrase by better emphasizing that the only novelty is that the patients are Korean.

Materials and methods:

Lines 86-88: What diagnostic criteria were used for IBM?

Line 93: I think it is better to rephrase with “a higher rate compared to other hereditary myopathies” as 15% specificity cannot be defined as a “high rate”.

Results:

Line 170: How was muscle weakness assessed? What is meant by muscle involvement pattern (there are no references to clinical scales in the methods)?

Lines 178-197: the paragraph on muscle MRI is confusing. I recommend only underlining the presence of common typical aspects and subsequently of the atypical/uncommon ones.

Lines 199-201: there are 2 antiCN1A+ patients with positive MSA, i.e. P5 and P8. Rather than percentages, I would report the raw data given the low sample size. In particular, P5 is positive for SRP with biopsy data (Table 1) consistent with the diagnosis of IMNM. Could it be a case of dual diagnosis (it would be interesting to know clinical and radiological information)? P8 is instead positive for both MDA5 and SAE1. What antibody titres does it have (borderline or high positive)?

Line 203: paragraph 3.3 is not of great interest as it is difficult to compare two groups of such different numbers and extremely heterogeneous within them (different forms of inflammatory myopathy in each group)

Line 231: Tawara et al (11 in text) and Lucchini et al (PMID 34068623) citations are missing.

Lines 232-233: rephrase because it seems that the data is confirmed in terms of percentage and not the fact that there are positive patients.

Lines 240-242: polymyositis cannot be considered a pre-stage of IBM. However, it is common not to find all the typical histological abnormalities in the muscle biopsy of IBM patients.

Lines 254-257: the meaning of the sentence is unclear.

Lines 260-261: There is a distinctive MRI pattern in the lower limbs (Tasca et al PMID 25808807).

Line 265: Be more specific about the adductor definition (there are different adductor muscles).

Lines 287-291: Explain that VCP myopathy is an inherited form of IBM. I would also be more cautious about the rare finding of anti-CN1A antibodies in patients with VCP myopathy.

Line 292: Have validated/standardized clinical scales been used?

Figures:

Legend for figure 3: correct the association patient-figure (P1 and P6 both F-K)

Figure 1: I suggest to comment in the text that high titer of anti-CN1A antibodies is almost exclusive to IBM (no patients with other IIM with ratio >3)

6. PLOS authors have the option to publish the peer review history of their article (what does this mean?). If published, this will include your full peer review and any attached files.

Reviewer #1: No

---

## [Author Response · Author response to Decision Letter 0]

15 Mar 2023

Mar 15th, 2023

Dear Editor,

Thank you for your letter regarding our submitted manuscript and reviewers’ comments. 

We have made the necessary corrections in accordance with the suggestions of the editior and reviewer, and detailed replies to the reviewers how we have responded to the points raised by the referee. Also, in the revised submit manuscript, we have highlighted all changes in yellow within our revised manuscript. 

We believe that the comments have improved the quality of our manuscript, and we hope you will find our revised manuscript acceptable for publication.

Sincerely Yours,

Hyung Jun Park

Hyung Jun Park, MD, PhD, 

Department of Neurology, Gangnam Severance Hospital, Yonsei University College of Medicine, 211 Eonju-ro, Gangnam-gu, Seoul 135-720, Korea 

Tel: +82-2-2019-3329

Fax: +82-2-3462-5904

E-mail: hjpark316@yuhs.ac

Point-by-point Response to Reviewers’ Comments

Reviewer #1

Comment 1: Abstract :Lines 42-43: in the abstract the authors presented only one group (inflammatory myopathies). The expression "In the inflammatory myopathy group" is therefore superfluous.

Response 1: Thank you for the comment. We removed unnecessary parts as suggested.

Line 42 – 45: Anti-NT5c1A antibody was most frequently identified in patients with inclusion body myositis (IBM) (8/20, 40%), followed by dermatomyositis (2/13, 15.4%), immune-mediated necrotizing myopathy (2/28, 7.1%), and polymyositis (1/42, 2.4%).

Comment 2: Line 51: “no clinically significant difference”. What are the clinical scales used?

Response 2: Thank you for the comment, we added the clinical aspects to compare two groups as follows

Line 51 – 54 : Clinically significant differences were not found between anti-NT5c1A antibody-seropositive and seronegative IBM groups with respect to gender, age at symptom onset, age at diagnosis, disease duration, serum CK values, presence of other autoantibodies, dysphagia, and the pattern of muscle impairment.

Comment 3: Introduction: Lines 75-77: “to date, no other studies….in Korea”. Rephrase by better emphasizing that the only novelty is that the patients are Korean.

Response 3: Thank you for your advice, we repharased the sentence as follows

Line 77 – 79 : In Korea, there have been no previous studies on anti-NT5c1A autoantibodies and their clinical associations with inflammatory myopathy, including IBM.

Comment 4: Materials and methods: Lines 86-88: What diagnostic criteria were used for IBM?

Response 4: Thank you for the comment. For IBM, we also used the new criteria proposed in the EULAR/ACR sub-classification among inflammatory myopathies. To clarify, we amended the sentence as follows. 

Line 88 – 91 : IBM group was defined by new criteria proposed in the EULAR/ACR sub-classification tree based on the clinical phenotypes of finger flexor weakness and no response to treatment or by the status of rimmed vacuoles in histopathologic findings [1].

Comment 5: Line 93: I think it is better to rephrase with “a higher rate compared to other hereditary myopathies” as 15% specificity cannot be defined as a “high rate”.

Response 5: Thank you for the proper comment. We revised the phrase as you recommended.

Line 96 – 98 : a higher rate compared to other hereditary myopathies of anti-NT5c1A antibody seropositivity was detected in patients with valosin-containing protein (VCP)-related myopathy [13].

Comment 6: Results: Line 170: How was muscle weakness assessed? What is meant by muscle involvement pattern (there are no references to clinical scales in the methods)?

Response 6: Thank you for the critical point. The method and grading score system has been added in Method part. Also, we defined the meaning of muscle impairment pattern. Added parts are as follows. 

Line 111 – 116 : Manual muscle testing was performed and was graded using the Medial Medical Research Council (MRC) 5-point scale (0, 1, 2, 3-, 3, 3+, 4-, 4, 4+, 5-, and 5). The MRC grading included shoulder abduction, elbow extension, elbow flexion, wrist extension, wrist flexion, hip flexion, knee extension, knee flexion, ankle dorsiflexion, ankle plantarflexion, and neck flexion. Especially for IBM group, muscle impairment pattern was checked whether there was muscle weakness in the knee extensors and finger flexors compared to adjacent muscles.

Comment 7

Lines 178-197: the paragraph on muscle MRI is confusing. I recommend only underlining the presence of common typical aspects and subsequently of the atypical/uncommon ones.

Response 7: Thank you for the important comment. We have amended the corresponding sentence as follows.

Line 194 – 208: In T1-weighted lower-limb MR images, muscle atrophy and moderate-to-severe fatty replacement of the quadriceps muscles were found in three patients (P1, P2, and P7) (Fig 3). At the pelvis level (A, G, M, and S), the iliopsoas muscles (*) were almost normal even at advanced stages without any inflammatory changes (D, J, P, and V). At the thigh level (B, H, N, and T), muscle atrophy and moderate-to-severe fatty replacement of the quadriceps(arrowheads) muscles were observed in three patients (B, N, and T). At the calf level (C, I, O, and U), the medial gastrocnemius (white arrows) muscles were involved at the initial stage and the fatty replacement of tibialis anterior, extensor digitorum, and peroneus muscles at the advanced stage (O and U). However, the adductor longus and magnus muscles showed more prominent fatty replacement than did the quadriceps muscles in P6 patients (H). 

Short tau inversion recovery (STIR) signal increase was patchy and asymmetric in the thigh and calf muscles. At the pelvis level (D, J, P, and V), there were not high-signal changes in four patients. At the thigh level (E, K, Q, and W), hypersignals of the vastus lateralis, medialis, and intermedius muscles were identified in three patients (P1, P2, and P7). At the calf level, the anterior calf muscles showed no signal changes during the early stages (F and L).

Comment 8 Lines 199-201: there are 2 antiCN1A+ patients with positive MSA, i.e. P5 and P8. Rather than percentages, I would report the raw data given the low sample size. In particular, P5 is positive for SRP with biopsy data (Table 1) consistent with the diagnosis of IMNM. Could it be a case of dual diagnosis (it would be interesting to know clinical and radiological information)? P8 is instead positive for both MDA5 and SAE1. What antibody titres does it have (borderline or high positive)?

Response 8: Thank you for the critical advice. Firstly, we agreed with your opinion it is not appropriate to indicate “percentages” because of low sample size. We have reported the raw number of patients (Line 209 – 212). Secondly, when we classify the patients into subgroups, EULAR/ACR classification tree was used as we mentioned “method” section. Accoring to the classification tree, it is preferentially divided into PM and IBM due to clnical features (finger flexor weakness and response to treatment). After classifying IBM, to SRP or HMGCR autoantibodies were used to classify PM and IMNM. Therefore, according to our operational definition, P5 is diagnosed as IBM. Thirdly, we have indicated all band positivity in table 1. 

Line 209 – 212: Testing for myositis-related autoantibodies in the IBM group revealed reactivity against Ro-52 in 4 patients. Reactivity against SRP, MDA5, SAE1, and Mi-2β were identified once each. Anti-SRP (P5), anti-MDA5 (P8), and anti-SAE1(P8) autoantibodies were found only in the anti-NT5c1A antibody seropositive group with IBM (Table 1 and S1 Table). 

Comment 9: Line 203: paragraph 3.3 is not of great interest as it is difficult to compare two groups of such different numbers and extremely heterogeneous within them (different forms of inflammatory myopathy in each group)

Response 9: Thank you for proper comment. We agree that comparing IBM and non-IBM inflammatory myopathies has less significance. However, authours also want to emphasize that anti-NT5c1A antibody can be identified in non-IBM groups and this itself does not have an absolute meaning. In addition, we expect that this data can be used as a comparative data when anti-NT5c1A antibodies are found in inflammatory myopathies other than IBM. We took your advice and added following sentences to avoid overinterpretation of our data in the discussion. 

Lines 255 – 256: However, it needs to be interpreted taking into account the small number and heterogeneity of non-IBM group.

Comment 10 Line 231: Tawara et al (11 in text) and Lucchini et al (PMID 34068623) citations are missing.

Response 10 Thank you for the comment. We have cited “Tawara et al” and added a reference (Lucchini et al for Ref 19) as you adviced. 

Lines 239 – 241: When only the results of previous studies using the ELISA method were compared, the sensitivity for IBM diagnosis was almost similar [4, 9, 11, 19].

Comment 11

Lines 232 – 233: rephrase because it seems that the data is confirmed in terms of percentage and not the fact that there are positive patients.

Response 11 : Thank you for the comment. We have rephrased the sentence as follows.

Lines 242 – 243: We also confirmed that anti-NT5c1A antibodies were identified in the patients with dermatomyositis (2/13), IMNM (2/28), and polymyositis (1/42).

Comment 12 Lines 240-242: polymyositis cannot be considered a pre-stage of IBM. However, it is common not to find all the typical histological abnormalities in the muscle biopsy of IBM patients.

Response 12: Thank you for the critical comment. We have amended the corresponding sentence as follows.

Line 249 – 251: In addition, typical histological abnormalities of IBM are not in all muscle pathologies of patients. Many IBM patients have been initially misdiagnosed with polymyositis [20]. 

Comment 13 Lines 254 – 257: the meaning of the sentence is unclear.

Response 13: Thank you for the comment. We have clarified the meaning, and it has been modified as follows.

Lines 264 –268: Several studies demonstrated no relevance between anti-NT5c1A antibody seropositivity and the presence of rimmed vacuoles [5, 11-13], consistent with our findings. However, there is also a report that anti-NT5c1A antibody seropositive IBM group represented with a lower frequency of rimmed vacuoles than seronegative IBM group [10].

Comment 13 Lines 260-261: There is a distinctive MRI pattern in the lower limbs (Tasca et al PMID 25808807).

Response 13 Thank you so much for the sincere comment. We reviewed the recommended journal and added it as a reference 22, and the sentence was specified as follows.

Lines 271 – 273: Muscle MRI findings in IBM are known to have preferential involvement, mainly in the flexor digitorum profundus, anterior compartment of distal thigh muscles, and medial gastrocnemius, while the tibialis posterior, soleus, and pelvic muscles are known to be relatively preserved [20-22].

Comment 14 Line 265: Be more specific about the adductor definition (there are different adductor muscles).

Response 14 : Thank you for the proper comment. We have descripted specific adductor muscles as you recommended.

Lines 275 – 278: One patient had remarkable fat infiltration in the bilateral adductor muscles (adductor longs and magnus) with mild atrophy in the rectus femoris muscles. This finding is contrary to a previous study, in which the adductor longus and adductor magnus were most frequently spared in IBM [21].

Comment 15 Lines 287-291: Explain that VCP myopathy is an inherited form of IBM. I would also be more cautious about the rare finding of anti-CN1A antibodies in patients with VCP myopathy.

Response 15 Thank you for the comment. We did not mean that VCP myopathy is an inherited form of IBM. The study was conducted based on the previous reports that the anti-NT5c1A antibody was identified in VCP myopathy among non-inflammatory myopathies. In order to avoid direct correlation between two, we have corrected the relevant parts as follows.

Line 298 – 304: There was a previous study to represent that the seropositivity for anti-NT5c1A antibody was higher among noninflammatory myopathies [13]. In addition, pathogenic variants of VCP have been identified in a small number of IBM patients [26, 27]. Along with previous findings, the result may support the possibility of a shared pathway with IBM and VCP-related myopathy in the production of anti-NT5c1A antibodies [13]. However, this is a still a rare finding and requires caution in interpretation. To date, no association has been established between VCP-related myopathy, anti-NT5c1A antibody, and IBM.

Comment 16 Line 292: Have validated/standardized clinical scales been used?

Response 16 Thank you for the critical comment. We have added following sentences. 

Line 307 – 309: Muscle weakness was evaluated with the MRC grade in our study. Therefore, further prospective studies are needed using validated/standardized clinical scales such as the adult myopathy assessment tool.

Comment 17 Figures: Legend for figure 3: correct the association patient-figure (P1 and P6 both F-K)

Response 17 Thank you for the comment. Figure legend have been corrected as follows.

Line 459: P2(A-F), P6(G-L), P7(M-R), and P1 (S-X)

Comment 18 Figure 1: I suggest to comment in the text that high titer of anti-CN1A antibodies is almost exclusive to IBM (no patients with other IIM with ratio >3)

Response 18 Thank you for the great comment. As you commented, we have added following sentence. 

Line 168 – 169:

Relatively high titer of anti-NT5c1A antibody (OD ratio≥3) was only confirmed in the IBM group.

Journal Requirements

Comment 1 Please ensure that your manuscript meets PLOS ONE's style requirements, including those for file naming. The PLOS ONE style templates can be found at https://journals.plos.org/plosone/s/file?id=wjVg/PLOSOne_formatting_sample_main_body.pdf and

Response 1 Thank you for the comment. we confirmed that our manuscript meets PLOS ONE’s style requirements

Comment 2 

Thank you for stating the following financial disclosure:“This research was supported by the Basic Science Research Program through the National Research Foundation of Korea (NRF) funded by the Ministry of Education (grant umber:2020R1I1A1A01068066)”

Response 2 Thank you for the comment. We added following part as you recommended. In addition, there were is one more funding source besides NRF. We have indicated on manuscript for “new faculty research seed money grant”. We also have added on Editorial manager.

Line 334 – 335: The funders had no role in study design, data collection and analysis, decision to publish, or preparation of the manuscript.

Comment 3 

In your Data Availability statement, you have not specified where the minimal data set underlying the results described in your manuscript can be found. PLOS defines a study's minimal data set as the underlying data used to reach the conclusions drawn in the manuscript and any additional data required to replicate the reported study findings in their entirety. All PLOS journals require that the minimal data set be made fully available. For more information about our data policy, please see http://journals.plos.org/plosone/s/data-availability. Upon re-submitting your revised manuscript, please upload your study’s minimal underlying data set as either Supporting Information files or to a stable, public repository and include the relevant URLs, DOIs, or accession numbers within your revised cover letter. For a list of acceptable repositories, please see http://journals.plos.org/plosone/s/data-availability#loc-recommended-repositories. Any potentially identifying patient information must be fully anonymized.

Response 3 Thank you for the comment. According to your advice, we have added the minimal data set as Table 1 (Line 429) and decripted in the first part of “Result”.

Line 157 – 162:

3.1 Demographic data

Table 1 summarize the demographic finding of 103 inflammatory myopathy patients. There were 40 males (39%) and 63 females (61%). The median ages at symptom onset and diagnosis were 55 (IQR: 44–63 years) and 57 (IQR: 57–65) years. The median disease duration was 10 (IQR: 3–32) months. Median serum CK levels were 2,777 (IQR: 545–7,231), the highest at 7,171 (IQR: 4,214–11,310) in IMNM, followed by polymyositis, dermatomyositis, and IBM. 

Comment 4 Your ethics statement should only appear in the Methods section of your manuscript. If your ethics statement is written in any section besides the Methods, please move it to the Methods section and delete it from any other section. Please ensure that your ethics statement is included in your manuscript, as the ethics statement entered into the online submission form will not be published alongside your manuscript.

Comment 4 Thank you for the comment. We have removed the ethis statement besides the Medthods.

Comment 5

Response 5 : Thank you for the comment. We confirmed the reference list.

---

## [Decision Letter · Decision Letter 1]

29 Mar 2023

Clinical significance of anti-NT5c1A autoantibody in Korean patients with inflammatory myopathies

PONE-D-22-33685R1

Dear Dr. 

We’re pleased to inform you that your manuscript has been judged scientifically suitable for publication and will be formally accepted for publication once it meets all outstanding technical requirements.

Kind regards,

Latika Gupta

Academic Editor

PLOS ONE

Additional Editor Comments (optional):

-

Reviewers' comments:

Reviewer's Responses to Questions

**Comments to the Author**

1. If the authors have adequately addressed your comments raised in a previous round of review and you feel that this manuscript is now acceptable for publication, you may indicate that here to bypass the “Comments to the Author” section, enter your conflict of interest statement in the “Confidential to Editor” section, and submit your "Accept" recommendation.

Reviewer #1: All comments have been addressed

2. Is the manuscript technically sound, and do the data support the conclusions?

Reviewer #1: Yes

3. Has the statistical analysis been performed appropriately and rigorously? 

Reviewer #1: Yes

4. Have the authors made all data underlying the findings in their manuscript fully available?

Reviewer #1: Yes

5. Is the manuscript presented in an intelligible fashion and written in standard English?

Reviewer #1: Yes

6. Review Comments to the Author

Reviewer #1: The authors properly addressed to the Reviewer's suggestions improving the quality of the manuscript.

7. PLOS authors have the option to publish the peer review history of their article (what does this mean?). If published, this will include your full peer review and any attached files.

Reviewer #1: No

---

## [Editor Report · Acceptance letter]

5 Apr 2023

PONE-D-22-33685R1 

Clinical significance of anti-NT5c1A autoantibody in Korean patients with inflammatory myopathies 

Dear Dr. Lee:

I'm pleased to inform you that your manuscript has been deemed suitable for publication in PLOS ONE. Congratulations! Your manuscript is now with our production department. 

Kind regards, 

on behalf of

Dr. Latika Gupta 

Academic Editor

PLOS ONE